# HANDLING COST AND CONSTRAINTS WITH OFF-POLICY DEEP REINFORCEMENT LEARNING

## ABSTRACT

Methods for off-policy deep reinforcement learning (DRL) offer improved sample efficiency relative to their on-policy counterparts, due to their ability to reuse data throughout the training process. For continuous action spaces, the most popular approaches to off-policy learning include policy improvement steps where a learned state-action ($Q$) value function is maximized over selected batches of data. These updates are often paired with regularization to combat associated overestimation of $Q$ values. With an eye toward safety, we revisit this strategy in environments with "mixed-sign" reward functions; that is, with reward functions that include independent positive (incentive) and negative (cost) terms. This setting is common in real-world applications, and may be addressed with or without constraints on the cost terms. We find the combination of function approximation and a term that maximizes $Q$ in the policy update to be problematic in such environments, because systematic errors in value estimation impact the contributions from the competing terms asymmetrically. This results in overemphasis of either incentives or costs and may severely limit learning. We explore two remedies to this issue. First, consistent with prior work, we find that periodic resetting of $Q$ and policy networks can be used to reduce value estimation error and improve learning in this setting. Second, we formulate novel off-policy actor-critic methods for both unconstrained and constrained learning that do not maximize $Q$ in the policy update. We find that this second approach, when applied to continuous action spaces with mixed-sign rewards, consistently and significantly outperforms state-of-the-art methods augmented by resetting. We further explore the applicability of our approach to more frequently-studied control problems that do not have mixed-sign rewards, finding it to both more reliably produce competent performance and be competitive in terms of overall performance.

## 1 INTRODUCTION

Model-free deep reinforcement learning (DRL) algorithms have shown significant potential in numerous domains, from robotic manipulation (Akkaya et al., 2019) to complex games (Mnih et al., 2015) to plasma control (Degrave et al., 2022). To become more widely used in real-world applications, however, further improvements in both training efficiency and agent safety are necessary.

Improved sample efficiency of DRL reduces training costs, whether through computation or runtime on physical systems. Off-policy methods for DRL reuse data collected throughout the training process, offering sample efficiency superior to that of on-policy approaches. State-of-the-art off-policy approaches typically leverage a replay buffer of experiences, learn a state-action value ($Q$) function, and find a policy that maximizes the learned $Q$ function. When applied to continuous action spaces, either a stochastic (Haarnoja et al., 2019) or deterministic (Fujimoto et al., 2018) policy representation may be used. Various approaches to regularizing learning in this setting have enabled significant increases in efficiency by enabling more network updates to be made per step of training data collected (Li et al., 2023; D'Oro et al., 2023).

Beyond training requirements, agents intended for real-world use must be safe. Environments where safety is critical typically include competing positive (incentive) and negative (cost) terms in their reward functions. The two may be considered together, or separately through constraints on the cost terms. In this work, we observe deficient behavior of state-of-the-art off-policy approaches for

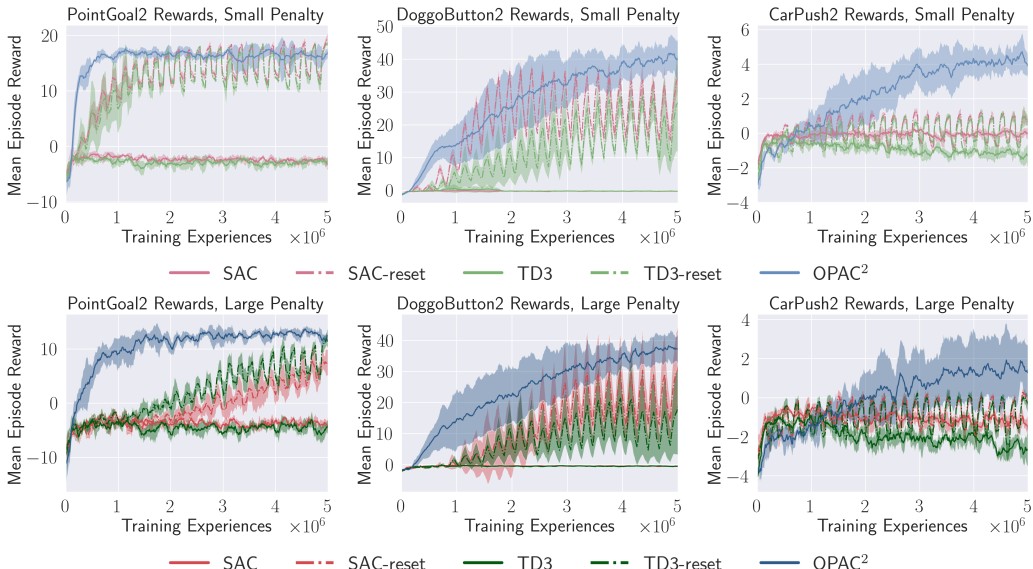

Figure 1: Sample unconstrained Learning on Safety Gym. **Top row**: learning with relatively small cost weights. **Bottom row**: learning with larger cost weights. SAC and TD3 struggle to learn without resetting, particularly as the weight on cost terms increases. Even with periodic resets, they do not match the performance of OPAC$^2$.

continuous control in such "mixed-sign" environments. This scenario occurs in numerous application spaces, from robotics (navigation in the presence of obstacles, robotic surgery) to resource allocation (when both performance and efficiency must be considered), to financial decision-making (where the risk of losses exists in the pursuit of gains).

Our contributions in this work are the following:

- We diagnose the cause of the aforementioned poor performance, finding that it stems from value estimation error impacting contributions from different reward terms asymmetrically.

- To address the issue, we provide a novel algorithm building around the off-policy actor-critic originally proposed by Degris et al. (2012). We empirically demonstrate that our method is not prone to harmful levels of value estimation error, and that it provides effective learning in both the unconstrained and constrained settings.

We find that in environments with mixed-sign rewards, our method significantly outperforms existing approaches where $Q$ is maximized in the policy update (Haarnoja et al., 2019; Fujimoto et al., 2018), even when they are augmented by resetting (Nikishin et al., 2022). We further find that our algorithm is competitive with these approaches, as well as more reliable in the sense of producing at least moderate competence, on tasks that do not include mixed-sign rewards.

## 2 PRELIMINARIES: OFF-POLICY RL FOR CONTINUOUS ACTION SPACES

Reinforcement learning (RL) considers the problem of maximizing discounted returns in a Markov Decision Process (MDP) $(\mathcal{S}, \mathcal{A}, \mathcal{P}, r, \gamma)$, where $\mathcal{S}$ is the state space, $\mathcal{A}$ is the action space, $\mathcal{P}$ are the transition dynamics, $r$ is the reward function, and $\gamma \in [0, 1)$ is the discount factor. RL maximizes the value function $V(\mathbf{s}_t) := \sum_{t'=t}^{\infty} \mathbb{E}_\pi[\gamma^{t'-t} r(\mathbf{s}_{t'}, \mathbf{a}_{t'})|\mathbf{s_t}]$ at initial ($t = 0$) states for trajectories encountered by an agent with policy $\pi(\mathbf{a}_t|\mathbf{s}_t)$ in an environment. The state-action value function $Q(\mathbf{s}_t, \mathbf{a}_t) := \sum_{t'=t}^{\infty} \mathbb{E}_\pi[\gamma^{t'-t} r(\mathbf{s}_{t'}, \mathbf{a}_{t'})|\mathbf{s_t}, \mathbf{a}_t]$ is the expected future discounted reward from a state $\mathbf{s_t}$, given that action $\mathbf{a}_t$ is taken at time $t$.

Practical approaches to off-policy deep reinforcement learning maintain a replay buffer $\mathcal{D}$ of transition tuples $(\mathbf{s}, \mathbf{a}, r, \mathbf{s}', d)$, where $d$ indicates episode termination, and interleave data collection

with network updates based on samples from the buffer. The network updates include both policy evaluation and policy improvement steps.

Policy evaluation typically involves estimation of the $Q$ function via a Bellman backup with loss

$$L(\phi) = \mathbb{E}_{(\mathbf{s},\mathbf{a},r,\mathbf{s}',d)\sim\mathcal{D}} \left[ \left( r + \gamma(1-d)Q^\pi_{\text{targ}}(\mathbf{s}',\mathbf{a}') - Q^\pi(\mathbf{s},\mathbf{a}) \right)^2 \right], \tag{1}$$

where $\mathbf{a}' \sim \pi_\theta(\mathbf{a}|\mathbf{s}')$ is sampled from the current policy evaluated at the next state $\mathbf{s}'$. A target network $Q^\pi_{\text{targ}}$, delayed from the current estimate for $Q^\pi$, is often used to promote learning stability (Mnih et al., 2015).

Policy improvement aims to maximize expected reward. The difference between the density of states encountered by the current policy and the totality of the replay buffer is often ignored in this step, introducing a bias in the policy estimate that is often manageable (Fu et al., 2019). In practice this amounts to learning a more general policy, applicable to the full range of scenarios encountered by the agent throughout the training process. The RL objective, for policy parameterized by variables $\theta$, may then be written as

$$J(\theta) \approx E_{\mathbf{s}\sim\mathcal{D}}V^{\pi_\theta}(\mathbf{s}) = E_{\mathbf{s}\sim\mathcal{D},\mathbf{a}\sim\pi_\theta(\mathbf{a}|\mathbf{s})}Q^{\pi_\theta}(\mathbf{s},\mathbf{a}) \tag{2}$$

The chain rule may be used to compute a policy gradient for this expression, and a more tractable form obtained by ignoring the second term:

$$\begin{aligned}
\nabla_\theta J(\theta) &\approx E_{\mathbf{s}\sim\mathcal{D},\mathbf{a}\sim\pi_\theta(\mathbf{a}|\mathbf{s})} \left[ \nabla_\theta \log(\pi_\theta(\mathbf{a}|\mathbf{s}))Q^{\pi_\theta}(\mathbf{s},\mathbf{a}) + \nabla_\theta Q^{\pi_\theta}(\mathbf{s},\mathbf{a}) \right] \\
&\approx E_{\mathbf{s}\sim\mathcal{D},\mathbf{a}\sim\pi_\theta(\mathbf{a}|\mathbf{s})} \left[ \nabla_\theta \log(\pi_\theta(\mathbf{a}|\mathbf{s}))Q^{\pi_\theta}(\mathbf{s},\mathbf{a}) \right]
\end{aligned} \tag{3}$$

Neglect of the second term, which may be difficult to compute off-policy, is shown to preserve the local minima to which the policy converges in the tabular case in Degris et al. (2012). To reduce variance of the policy gradient estimate, Degris et al. (2012) recommend removing a state dependent baseline, resulting in

$$\nabla_\theta J(\theta) \approx E_{\mathbf{s}\sim\mathcal{D},\mathbf{a}\sim\pi_\theta(\mathbf{a}|\mathbf{s})} \left[ \nabla_\theta \log(\pi_\theta(\mathbf{a}|\mathbf{s}))A^{\pi_\theta}(\mathbf{s},\mathbf{a}) \right], \tag{4}$$

where $A^{\pi_\theta}(\mathbf{s},\mathbf{a}) = Q^{\pi_\theta}(\mathbf{s},\mathbf{a}) - V^{\pi_\theta}(\mathbf{s})$ is the advantage function.

There are two common alternatives to (Eq. 3), both more commonly used today. The first is the deterministic policy gradient (DPG; Silver et al. (2014)), which represents the policy as a deterministic function $\mu_\theta(\mathbf{s})$. With that representation, the policy may be updated according to

$$\nabla_\theta J(\theta) \approx \mathbb{E}_{\mathbf{s}\sim\mathcal{D}} \left[ \nabla_\theta \mu_\theta(\mathbf{s})\nabla_\mathbf{a} Q^{\mu_\theta}(\mathbf{s},\mathbf{a})|_{\mathbf{a}=\mu_\theta(\mathbf{s})} \right] \tag{5}$$

The second is the reparameterization trick, which encodes the stochasticity of the policy in an independent noise variable. The resulting form has provably lower variance than (3) (Kingma et al., 2015), and is used by Soft Actor-Critic (SAC; Haarnoja et al. (2018; 2019)). SAC also uses a "squashed" action representation for enforcing control bounds: $\mathbf{a}_\theta(\mathbf{s},\xi) = \tanh(\mu_\theta(\mathbf{s})+\sigma_\theta(\mathbf{s})\odot\xi)$, where $\xi \sim \mathcal{N}(\mathbf{0},I)$ and $\mu_\theta, \sigma_\theta$ are the state-dependent mean and variance, respectively. Including a second term to encourage policy entropy, SAC performs soft policy improvement via

$$\nabla_\theta J(\theta) = \mathbb{E}_{\mathbf{s}\sim\mathcal{D},\xi\sim\mathcal{N}(\mathbf{0},\mathbf{1})} \left[ \nabla_\theta Q^{\pi_\theta}(\mathbf{s},\mathbf{a}_\theta(\mathbf{s},\xi)) - \alpha\nabla_\theta \log(\pi_\theta(\mathbf{a}_\theta(\mathbf{s},\xi)|\mathbf{s})) \right]. \tag{6}$$

Modern off-policy approaches using the deterministic policy gradient (e.g., TD3; Fujimoto et al. (2018)) and reparameterization trick (e.g., SAC) consider multiple $Q$ functions to combat overestimation of $Q$ originating in its maximization in the policy update steps (5) and (6). This mechanism was addressed for discrete action spaces in Hasselt (2010) and explained in detail for continuous action spaces in Fujimoto et al. (2018). It is rooted in overestimation errors in the value update being propagated by the policy update. To combat this effect, both TD3 and SAC employ the "clipped double $Q$ trick" in the policy evaluation step

$$L(\phi) = \mathbb{E}_{(\mathbf{s},\mathbf{a},r,\mathbf{s}',d)\sim\mathcal{D}} \left[ \left( r + \gamma(1-d)\min_{i=1,2} Q^\pi_{\phi_i,\text{targ}}(\mathbf{s}',\mathbf{a}') - Q^\pi_\phi(\mathbf{s},\mathbf{a}) \right)^2 \right] \tag{7}$$

to train two $Q$ functions with different initial parameters $\phi_1, \phi_2$ toward the minimum of their outputs, for each sampled experience. SAC also uses the minimum of the two $Q$ functions in the policy update. Other methods (Chen et al., 2021; Hiraoka et al., 2022) go further, training larger ensembles

of $Q$ functions. These adjustments mitigate $Q$ overestimation, but do not fully solve the problem. In some cases, $Q$ may still be overestimated; in others, it may be underestimated.

Notably, we find that these corrections are unnecessary with the off-policy actor-critic (Eq. 3). While $Q$ may still be slightly overestimated, we find that the "gentler" updates provided by the off-policy actor-critic remove the tendency for large inaccuracies. The addition of the value function in Eq. 4 both provides variance reduction and weakens the feedback between the policy and Q updates, leading to learning that we find to be both reliable and performant.

## 3 THE IMPACT OF COMPETING OBJECTIVES

As explained in Fujimoto et al. (2018), the tendency of $Q$ to be systematically overestimated when maximized in the policy update occurs because of the pairing of that update with the use of $Q$ (albeit delayed) in the target of its own update (Eq. 1). SAC and TD3 combat this upward bias by learning two $Q$ functions, using each as the target for the other's update. To make sure this target is never a worse overestimate than just using the same $Q$ function, the target is set to the minimum of the two $Q$ functions (the "clipped double $Q$ trick"). SAC also uses this minimum in the policy update. Unfortunately, it is unlikely that any form of regularization will exactly balance the intrinsic tendency of $Q$ to be overestimated when maximized in the policy update.

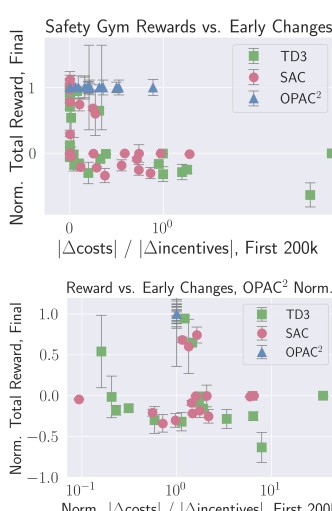

Figure 2: **Top:** Total (cost and incentive) reward per-episode of fully-trained agents as a function of cost adjustment relative to reward adjustment early in Safety Gym training. **Bottom:** SAC and TD3 perform best when prioritizing cost similarly to OPAC$^2$.

Now consider a continuous control problem where the reward function is a sum of multiple independent objectives. A special case of this is when rewards are "mixed-sign;" that is, contain independent incentive ($> 0$) and cost ($< 0$) terms. As recently explored (Nikishin et al., 2022; Li et al., 2023), off-policy DRL agents are prone to overemphasizing experiences collected early in training. In the context of multiple independent objectives, this corresponds to excessive focus on the reward terms most easily accessed by an agent of low capability. Fixation on some terms may lead to others being neglected entirely, as has been observed for entire *tasks* in multi-task learning (Yu et al., 2020; Liu et al., 2021). The situation is more likely to become acute with mixed-sign rewards; favoring a given reward term early in training may cause the agent to ignore terms designed to counterbalance it. Even if overemphasis of particular terms does not lead to wholesale neglect of others, it is likely to impact the relative consideration of different terms by the agent and thereby impact behavior.

To explore this issue, we evaluated the performance of different off-policy algorithms on the OpenAI Safety Gym (Ray et al., 2019), a "mixed-sign" suite of robotic navigation tasks with obstacles. In the top panel of Figure 2, each marker represents 5 training runs on a given configuration, where a configuration is a combination of robot, obstacle set, task, and penalty assessed each time an obstacle is contacted. Each algorithm tackled the same set of 24 configurations, covering significant range in the ease of accessing incentive and cost terms early in training. The $x$-axis reflects how much an agent adjusts its cost levels relative to incentives early on: it is the *change* in average costs accumulated by the agent at the start of training compared to what it accumulates 200k steps in, divided by that same difference for incentives. Points with $x = 0$ were configured to have 0 penalty (ignore costs). The $y$-axis is the average total reward (including costs and incentives) per-episode of the fully trained agent, normalized by the level achieved by our updated off-policy actor-critic (OPAC$^2$) agent. We observe that SAC and TD3 are typically able to accumulate incentives well only when cost is close to ignored, and tend to prioritize cost more highly than OPAC$^2$. In the bottom panel, we plot all configurations with nonzero penalty weights. We normalize the x and y values for each point by those obtained by OPAC$^2$ in the environment configuration corresponding to the point. We observe that SAC and TD3 are able to compete with OPAC$^2$ only when they adjust cost and reward similarly to OPAC$^2$ early in training, suggesting improper balancing elsewhere. We provide more details on these experiments in Appendix A and describe OPAC$^2$ further below.

## 4    ADDRESSING ERRONEOUS $Q$ ESTIMATES

We investigate two routes for addressing this problem. Section 4.1 examines the efficacy of a popular regularization strategy in this setting. In Section 4.2, we propose a novel algorithm with desirable properties for this problem.

### 4.1    REGULARIZATION VIA RESETTING

One method recently shown to be effective for improving off-policy learning is to periodically reset policy and value networks throughout training, while preserving accumulated experience (Nikishin et al., 2022). This method has been shown to enable learning that is extremely sample-efficient, by allowing many network updates to be conducted per collected data point (learning at high "replay ratios"). Li et al. (2023) observed that resetting reduces temporal difference (TD) error on validation (i.e., not seen in training) transitions. We find this observation to be particularly relevant in mixed-sign environments; in many cases, resetting is seen to enable learning with SAC and TD3 when it would not otherwise be possible. However it is not seen to provide optimal performance, and could be inappropriate in scenarios where *cumulative* cost is a consideration[1].

### 4.2    UPDATING THE OFF-POLICY ACTOR-CRITIC

To more directly address erroneous $Q$ estimates in environments with competing objectives, we revisit the off-policy actor-critic (Eq. 4). We seek to build a practical algorithm around it, leveraging techniques from more recent methods as well as novel extensions to reduce the variance on the policy gradient estimate while leveraging its potentially less biased value estimation. Differing from Degris et al. (2012), we neglect importance weights in our gradient estimates, an approximation that significantly reduces variance and amounts to learning a more general policy over all states in the buffer (rather than one tailored to the current density of states). We adopted several aspects of Soft Actor-Critic, including $Q$ and $V$ updates that match the original version of SAC (though we require only a single $Q$ network) and "squashing" of actions with a hyperbolic tangent to respect control bounds. The latter tactic reduces policy gradient bias relative to methods that allow the environment to clip actions (Fujita & Maeda, 2018). Finally, we normalized advantage estimates prior to use in the policy gradient (Eq. 4).

Incentivizing policy entropy has been shown to improve learning in complex problems and with high-dimensional control spaces (Haarnoja et al., 2018). One existing approach is the "maximum-entropy" framework of Haarnoja et al. (2019), wherein entropy is bundled with the reward. While this strategy is compatible with our approach, we also considered the use of an entropy bonus to be added directly to the policy loss. Empirically, we found this entropy bonus to outperform the "max-entropy" strategy (Figure 3) in environments with mixed-sign rewards. To accommodate squashing, our bonus used an action sampled with the reparameterization trick and was optimized towards a target entropy level, similar to Haarnoja et al. (2019). Pseudocode for our unconstrained algorithm (OPAC[2]) is provided in Appendix B.

#### 4.2.1    CONSTRAINED APPROACH

Reinforcement learning with costs may alternatively be formulated with constraints. Differing from the unconstrained setting discussed above, Constrained Markov Decision Processes (CMDPs) have positive rewards $r(\mathbf{s}, \mathbf{a})$ and costs $c(\mathbf{s}, \mathbf{a})$ provided separately for each time step, as well as an overall constraint $C(\tau) = F(c(\mathbf{s}_1, \mathbf{a}_1), \ldots, c(\mathbf{s}_T, \mathbf{a}_T))$ defined over the whole trajectory $\tau \equiv \mathbf{s}_1, \mathbf{a}_1, \ldots, \mathbf{s}_T, \mathbf{a}_T$. The associated learning problem is to maximize the value function $V^\pi(\mathbf{s}_t)$ associated with rewards, such that the expected value of the constraint over trajectories sampled by the agent will not exceed a fixed threshold $M$: $J_C(\theta) = E_{\tau \sim p_\theta(\tau)} C(\tau) < M$. Here $p_\theta(\tau)$ is the probability distribution of trajectories $\tau$ encountered by an agent parameterized by $\theta$. While other functions are possible, here we will be concerned with constraints on total trajectory cost: $C(\tau) = \sum_{t=1}^{T} c(\mathbf{s}_t, \mathbf{a}_t)$.

Constrained RL is often conducted using dual methods (Bertsekas, 1996; Boyd & Vandenberghe, 2004), and has previously been explored off-policy with SAC (Ha et al., 2020; Zhou et al., 2022;

---

[1]D'Oro et al. (2023) incorporated offline training after resets to mitigate this issue.

Yang et al., 2021). The goal is to learn a Lagrange multiplier $\beta$ that scales cost relative to reward during policy optimization in order to satisfy the constraint. This circumvents the need for reward shaping, as the weight of the cost terms is learned. As $\beta$ changes throughout training, it is beneficial to train separate $Q$ and $V$ networks for reward ($Q_r$, $V_r$) and cost ($Q_c$, $V_c$). To facilitate cost matching in the states currently frequented by the agent, we update $\beta$ toward the average cost incurred by the agent *over only the last epoch of training*. This update, like those on other quantities, is conducted every time step. Our full constrained approach, Constrained Off-Policy Actor-Critic, SQUAshed and REgularizeD (C-OPAC$^2$), is given in Algorithm 1. This differs from the unconstrained approach in two ways: we now learn two $Q$ and $V$ networks, one each for rewards and costs, and we also learn $\beta$ in order to satisfy the constraint.

---

**Algorithm 1** Constrained Off-Policy Actor-Critic, SQUAshed and REgularizeD (C-OPAC$^2$)

---

1: **Input:** Initial policy parameters $\theta$; $Q_r$, $Q_c$ parameters $\phi_r$, $\phi_c$; $V_r$, $V_c$ parameters $\psi_r$, $\psi_c$
2: **Input:** Initial entropy weight $\alpha$, penalty weight $\beta$
3: Initialize $V_r$, $V_c$ target network parameters: $\psi_{r,\mathrm{targ}} \leftarrow \psi_r$; $\psi_{c,\mathrm{targ}} \leftarrow \psi_c$
4: Initialize replay buffer $\mathcal{D} = \emptyset$, a ring buffer of fixed size
5: **for** iteration $k \in [0, \ldots, K-1]$ **do**
6:     **for** step $s \in [0, \ldots, S-1]$ **do**           $\triangleright$ Typically take just one step ($S = 1$)
7:         Sample $\mathbf{a} \sim \pi_{\theta_k}(\mathbf{a}|\mathbf{s})$; observe $\mathbf{s}' \sim p(\mathbf{s}'|\mathbf{s}, \mathbf{a})$     $\triangleright$ One step in CMDP
8:         Store transition tuple $(\mathbf{s}, \mathbf{a}, \mathbf{s}', r(\mathbf{s}, \mathbf{a}), c(\mathbf{s}, \mathbf{a}))$ in $\mathcal{D}$
9:     **end for**
10:     **for** gradient step $g \in [0, \ldots, G-1]$ **do**
11:         Sample batch $B = \{(\mathbf{s}_i, \mathbf{a}_i, \mathbf{s}'_i, r_i, c_i, d_i)\}$ from $\mathcal{D}$
12:         Update $\beta$: $\beta \leftarrow \beta - \lambda_\beta \nabla_\beta \beta(M - J_C(\theta))$    $\triangleright$ $J_C(\theta)$ computed over most recent epoch
13:         Compute Q error: $\mathcal{E}_Q(B) = \sum_i [Q_r(\mathbf{s}_i, \mathbf{a}_i) - (r_i + \gamma(1 - d_i)V_{r,\mathrm{targ}}(\mathbf{s}_i))]^2 +$
14:                           $[Q_c(\mathbf{s}_i, \mathbf{a}_i) - (c_i + \gamma(1 - d_i)V_{c,\mathrm{targ}}(\mathbf{s}_i))]^2$
15:         Update Q: $\phi_r \leftarrow \phi_r - \lambda_\phi \nabla_{\phi_r} \mathcal{E}_Q(B)$; $\phi_c \leftarrow \phi_c - \lambda_\phi \nabla_{\phi_c} \mathcal{E}_Q(B)$
16:         Sample $\mathbf{a}_{i,\pi} \sim \pi(\mathbf{a}|\mathbf{s}_i)$, using $\tanh$ squashing
17:         Compute V error: $\mathcal{E}_V(B) = \sum_i [V_r(\mathbf{s}_i) - Q_r(\mathbf{s}_i, \mathbf{a}_{i,\pi})]^2 + [V_c(\mathbf{s}_i) - Q_r(\mathbf{s}_i, \mathbf{a}_{i,\pi})]^2$
18:         Update V: $\psi_r \leftarrow \psi_r - \lambda_\psi \nabla_{\psi_r} \mathcal{E}_V(B)$; $\psi_c \leftarrow \psi_c - \lambda_\psi \nabla_{\psi_c} \mathcal{E}_V(B)$
19:         Compute $A(\mathbf{s}_i, \mathbf{a}_{i,\pi}) = Q_r(\mathbf{s}_i, \mathbf{a}_{i,\pi}) - V_r(\mathbf{s}_i) - \beta[Q_c(\mathbf{s}_i, \mathbf{a}_{i,\pi}) - V_c(\mathbf{s}_i)]$   $\triangleright$ Normalize
20:         Sample $\mathbf{a}_{\mathrm{rp},\pi} \sim \pi(\mathbf{a}|\mathbf{s}_i)$ with reparameterization trick (for $\tanh$ squashing)
21:         Compute $\pi$ loss: $\mathcal{E}_\pi(B) = \sum_i [\alpha \log \pi_\theta(\mathbf{a}_{\mathrm{rp},\pi}|\mathbf{s}_i) - A(\mathbf{s}_i, \mathbf{a}_{i,\pi}) \log(\pi_\theta(\mathbf{a}_{i,\pi}|\mathbf{s}_i)]$
22:         Update $\pi$: $\theta \leftarrow \theta - \lambda_\theta \nabla_\theta \mathcal{E}_\pi(B)$
23:         Update $\alpha$ : $\alpha \leftarrow \alpha - \lambda_\alpha \nabla_\alpha [-\alpha \log(\pi(\mathbf{a}_{i,\pi}|\mathbf{s}_i)) - \alpha \mathcal{H}_{\mathrm{target}}]$
24:         Update value targets: $\psi_{r,\mathrm{targ}} \leftarrow \rho \psi_{r,\mathrm{targ}} + (1 - \rho)\psi_r$; $\psi_{c,\mathrm{targ}} \leftarrow \rho \psi_{c,\mathrm{targ}} + (1 - \rho)\psi_c$
25:     **end for**
26: **end for**

---

## 5 EXPERIMENTS

We used the OpenAI Safety Gym (Ray et al., 2019) to explore off-policy learning for environments with continuous action spaces and mixed-sign rewards. Safety Gym is a configurable set of robotic navigation tasks, wherein different robots must navigate through courses containing multiple obstacle types to perform different tasks as many times as possible in a fixed time window. The locations of goals and obstacles are randomized, leading to outcome variability and necessitating the learning of a generalized control strategy. For unconstrained experiments, each cost event incurred a fixed, negative reward. For constrained experiments, the weight coefficient for cost events was learned. We evaluated all robots and tasks for the most obstacle-rich (level 2) publicly available environments. Additional experimental details, including hyperparameters, are given in Appendix C.

### 5.1 UNCONSTRAINED LEARNING

To explore the effect of mixed-sign rewards on unconstrained off-policy learning, we considered both small and large static cost weights in our target environments. The larger penalty weights matched those used on-policy in Markowitz et al. (2023), while the smaller weights were reduced

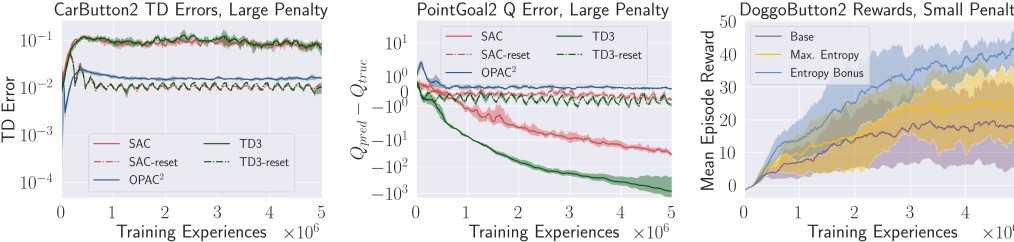

Figure 3: Unconstrained learning with mixed-sign rewards. **Left**: TD error in validation is generally higher with SAC and TD3 than OPAC$^2$, but may be reduced with resetting. **Middle**: $Q$ is underestimated by SAC and TD3 in all "Car" and "Point" environments, but may be improved by resetting. **Right**: Our entropy bonus (blue) outperforms max-entropy and no regularization when used with OPAC$^2$ on the higher-dimensional Doggo control problems.

by a factor of 2. We compared OPAC$^2$ with SAC and TD3, both with and without resets. A sample of the resulting learning curves is shown in Figure 1. The remainder, which follow similar qualitative trends, are provided in Appendix D. We find SAC and TD3 to be generally unable to reach adequate performance, particularly as the penalty weight increases. The situation is improved when resetting is applied; however, it often does not result in performance that reaches the level of OPAC$^2$. Attesting to the sample efficiency of off-policy methods, we find that OPAC$^2$ is typically able to reach performance comparable to that of Proximal Policy Optimization (PPO; Schulman et al. (2017b)) and Trust Region Policy Optimization (TRPO; Schulman et al. (2017a)) using 20–50 times fewer samples (based on results reported by Markowitz et al. (2023)). On the Doggo environments, OPAC$^2$ exceeds the positive reward accumulation of PPO and TRPO trained *without cost terms* and with twice as many samples (Ray et al., 2019).

To explore the issues affecting SAC and TD3, we examined the validation TD error as well as the $Q$ function error. As in Li et al. (2023), "validation" refers to a held-out set of transitions not in the replay buffer. Figure 3 and Appendix D demonstrate that SAC and TD3 are prone to large TD error, and that periodic resets mitigate this. This phenomenon was observed by Li et al. (2023) in the context of learning at high replay ratios. OPAC$^2$ is not nearly as prone to high TD error, and accordingly does not benefit from resets (Appendix D.1). We also compared $Q$ function estimates to discounted Monte Carlo returns to measure $Q$ function accuracy. We found that SAC and TD3 *underestimate* the true $Q$ value in all "Car" and "Point" environments, while the trend was more variable in the "Doggo" environments.

In the rightmost panel of Figure 3 and Appendix D.2, we display the efficacy of our entropy bonus strategy on "Doggo" environments. "Doggo" is a higher-dimensional control problem than "Car" and "Point" (12-dimensional vs. 2-dimensional), potentially leading it to require more exploration. In all environments with mixed-sign rewards tested, we observed better performance with our entropy bonus than with a maximum entropy approach.

## 5.2 CONSTRAINED LEARNING

We further compared all methods in the constrained setting, again on Safety Gym. In all environments, we chose a target cost level equal to half the cost accumulated by a fully-trained TRPO agent unaware of cost (as reported by Ray et al. (2019)). We chose this target to force the agent to strongly consider safely, but not be constrained to the point of being unable to complete the task. All agents learned separate value networks for reward and cost, to accommodate the variability of the penalty weight throughout training. We offer comparison with *constrained versions* of SAC and TD3.

A representative sample of constrained learning performance is given in Figure 4, with full results being provided in Appendix E. We found all methods capable of reaching the target cost level, with OPAC$^2$ achieving the highest positive rewards in all environments tested. Sample efficiency was again greatly improved relative to on-policy approaches, with performance matching that of state-of-the-art on-policy methods using roughly 50 times less data (Markowitz et al., 2023). TD3 and SAC were both seen to benefit from resetting, allowing them to reduce the error on their cost and

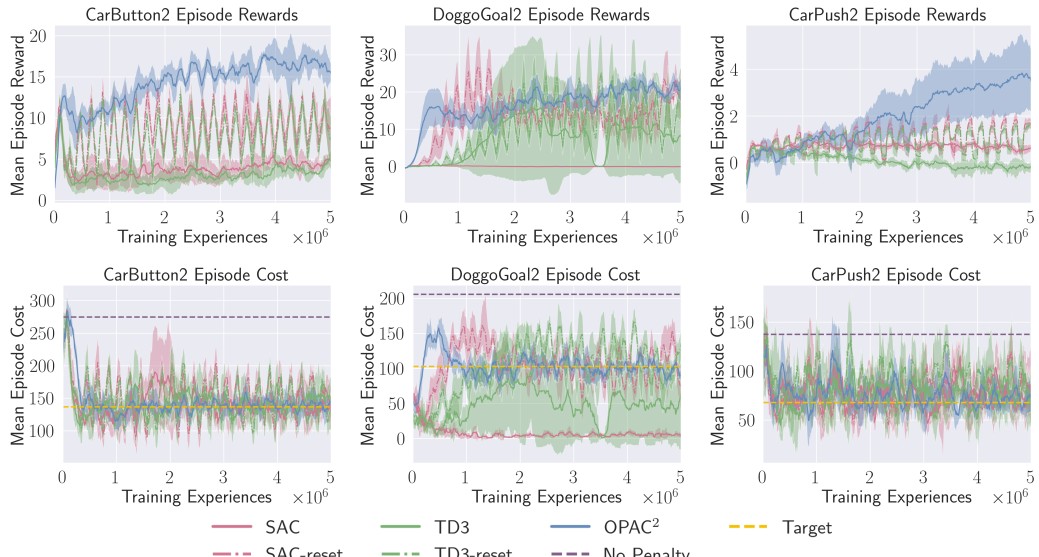

Figure 4: Representative sample of constrained learning on Safety Gym. **Top Row**: Positive reward accumulated by agents. Note that reward accumulation initially increases sharply while $\beta$, the penalty weight, is low. It dips and then rises again as $\beta$ increases and stabilizes, allowing performance to be optimized at the appropriate cost level. **Bottom Row**: Cost converges to the target level (yellow).

reward value estimates. However, this improvement is not always enough to match the performance of OPAC$^2$. Finally, we note that this approach is highly configurable; the initial value and learning rate of the cost weight $\beta$ may be adjusted upward to provide faster convergence to the prescribed cost target or downward to enable more early exploration and potentially higher final positive rewards.

### 5.3 EVALUATION ON DEEPMIND CONTROL

Given the consistently strong performance of OPAC$^2$ in Safety Gym, we chose to additionally evaluate it on 10 tasks without mixed-sign rewards from the DeepMind Control Suite (Tunyasuvunakool et al., 2020). We used the same 10 tasks as the authors of Nikishin et al. (2022), motivated by the fact that they provide some challenge for SAC. Following the practice of Agarwal et al. (2021), we aggregate performance across tasks using the interquartile mean (IQM), calculated as the mean score of the middle 50% of runs. We also computed performance profiles for each method, tabulating the fraction of experiments that exceeded each possible performance level.

As shown in Figure 5, the aggregated task performance of OPAC$^2$ is worse than SAC but on par with or slightly better than TD3 (full results in Appendix F). This is to be expected, given the likely lower variance of the policy gradient estimates of SAC. However, OPAC$^2$ was found to be more reliable than the other methods at producing at least some learning in each environment, as evidenced by the left side of the performance profile plot.

## 6 RELATED WORK

Our methods build on the off-policy actor-critic introduced in Degris et al. (2012), applied in differing forms over the past decade

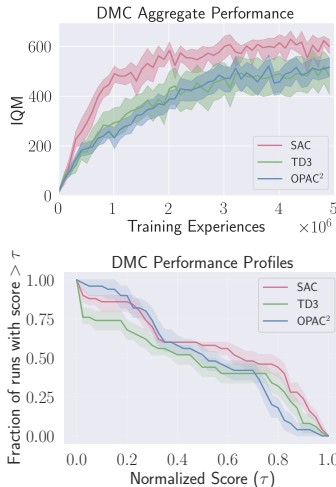

Figure 5: **Top**: Rewards aggregated over 10 tasks from the DeepMind Control Suite. **Bottom**: Frequency of reward accumulation matching given performance levels.

(Wang et al., 2017; Espeholt et al., 2018; Gu et al., 2017), and summarized in Levine et al. (2020). We compare it with more popular methods for off-policy DRL with continuous action spaces (Fujimoto et al., 2018; Haarnoja et al., 2019). The resetting strategy that we employ was explored by Nikishin et al. (2022) and subsequently further evaluated and extended (D'Oro et al., 2023; Schwarzer et al., 2023). Its ability to curtail TD error, as well as the role of TD error in limiting off-policy DRL more generally, was discussed in Li et al. (2023). To our knowledge, its impact on environments with mixed-sign rewards had not previously been quantified.

Constrained DRL has previously been explored on-policy (Bhatnagar, 2010; Achiam et al., 2017; Ray et al., 2019; Chow et al., 2019; Tessler et al., 2019; Paternain et al., 2019; Zhang et al., 2020; Markowitz et al., 2023; Moskovitz et al., 2023), typically leveraging dual gradient descent (Bertsekas, 1996; Boyd & Vandenberghe, 2004). Constraints have recently been applied off-policy (Ha et al., 2020; Zhou et al., 2022) for specific applications and with a distributional critic (Yang et al., 2021). They have also been applied to probabilistically ensure safety *during training* (Bharadhwaj et al., 2021), a useful requirement that we do not consider here. To our knowledge, this work is the first to apply constraints with the off-policy actor-critic, the first to explore the use of resetting in this context, and the first to specify constraints via consideration of only the most recent epoch in the replay buffer.

## 7 DISCUSSION

The success of OPAC$^2$ relative to SAC and TD3 in environments with mixed-sign rewards underscores the necessity of managing function approximation error in off-policy deep reinforcement learning. Off-policy DRL agents are known to be sensitive to experience gained early in training (Nikishin et al., 2022), tending to exploit any initial success they have. When multiple independent reward terms are present, this translates to some terms being favored over others. In environments with mixed-sign rewards, the explicit competition between incentives and costs exacerbates this tendency. This effect is fundamentally a result of overfitting in $Q$ estimation. While OPAC$^2$ does not fully eliminate $Q$ estimation errors, we empirically observe it to significantly dampen them. Further, the likely higher variance of the policy gradient estimate in OPAC$^2$ may actually be beneficial in this setting: while not being large enough to preclude learning, it may help to address competition between reward terms by preventing the policy from converging too quickly to one that ignores some terms. This combination gives OPAC$^2$ a significant advantage over methods that update the policy to maximize the $Q$ function estimate when competing reward terms are present, even when the latter are periodically "course corrected" by resetting.

Several additional observations may be made based on these findings. First, they argue for the use of a policy-gradient-based approach whenever mixed-sign rewards are present (including constrained learning). This is particularly true when resetting is impractical, for instance when cumulative cost is a consideration or for offline learning. When rewards do not include terms of mixed signs, we may expect OPAC$^2$ to perform competently and reliably, but often not as well as SAC. It is interesting to note that OPAC$^2$ showed strong learning on the `acrobot-swingup` task, the only DeepMind Control environment we tested that SAC and TD3 both failed to solve. Finally, we note that a policy-gradient-based approach may prove beneficial for multi-task or meta-learning, where competing reward terms from multiple tasks must be considered.

## 8 CONCLUSIONS

In this work, we examine the tendency of state-of-the-art approaches to off-policy DRL for continuous action spaces to struggle when applied to environments with mixed-sign rewards. We elucidate the role of function approximation error in the process; in particular, the sensitivity of off-policy methods to experience gathered early in training may lead to the neglect of some terms of the reward function. To remedy the situation, we consider both a popular regularization strategy (periodic resetting) and a novel approach that produces more reliable value estimates (OPAC$^2$). Empirically we find the latter, which represents an update to the off-policy actor critic (Degris et al., 2012) and related methods, to produce more performant learning in the presence of costs.

## ETHICS STATEMENT

In the interest of transparency and reproducibility, we provide full details necessary for reproducing experiments in the Appendix, as well as all source code. While our work is immediately targeted at making safer robotic agents, it is true that others could repurpose our code for malicious applications. We encourage the authors of any subsequent work to consider the societal impacts of future results.

## REPRODUCIBILITY STATEMENT

We provide detailed pseudocode of our constrained algorithm in Algorithm 1, and for our unconstrained algorithm in Algorithm 2. In Appendix C, we provide hyperparameters, evaluation procedures, and implementation details necessary for replicating all experiments. Additionally, we make the source code available here: (available in supplementary material while under review; will be posted publicly pending acceptance).

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

## A ADDITIONAL EMPIRICAL RESULTS ON THE IMPACT OF COMPETING OBJECTIVES

Here we supplement the empirical findings of Section 3. First, we note that the duration of "early" training reflected by the x-axis was chosen to match the time before the first network reset in experiments that featured network resetting.

In Figure 6, we show the breakdown of incentive and cost in the experiments that constitute Figure 2, again normalizing by the performance of OPAC$^2$ on each environment. We find that, particularly with nonzero cost weights, OPAC$^2$ achieves far higher levels of incentives than TD3 or SAC (left panel). OPAC$^2$ does accumulate higher costs than the others (right panel), but by not nearly as large a factor. These trends correspond to OPAC$^2$ accumulating much higher *total* reward (Figure 2), as is the goal of the optimization. When cost is considered in these environments, the SAC and TD3 agents largely fail to properly balance the competing objectives.

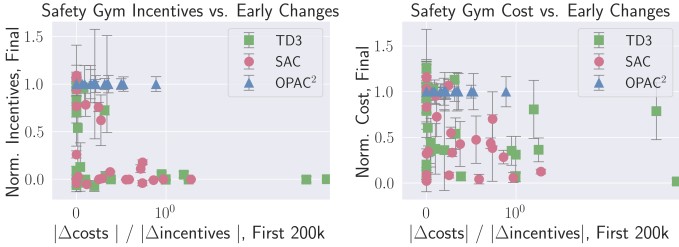

Figure 6: **Left panel**: Average positive reward (incentive) per episode at the end of training, as a function of initial prioritization of cost. **Right panel**: Average cost per episode at the end of training, as a function of initial prioritization of cost. In both cases, quantities are normalized by the OPAC$^2$ result for the given environmental configuration.

In Figure 7, we again show the incentives (left panel) and costs (right panel) at the end of training plotted against the ratio of early change in cost accumulation to early change in incentives gained, but this time normalized by the best on-policy result for an agent unaware of cost Ray et al. (2019). We see that OPAC$^2$ vastly outperforms the on-policy methods in terms of positive reward accumulation, despite using half as much training data. As OPAC$^2$ is increasingly configured to consider cost, its accumulation of both cost and reward decreases.

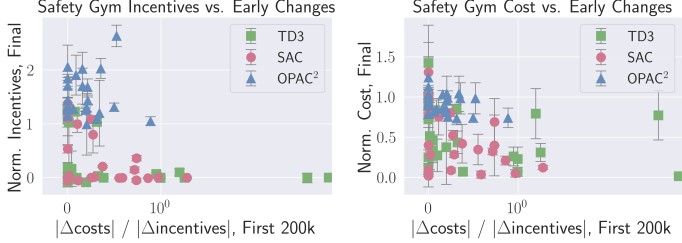

Figure 7: **Left panel**: Average positive reward (incentive) per episode at the end of training, as a function of initial prioritization of cost. **Right panel**: Average cost per episode at the end of training, as a function of initial prioritization of cost. In both cases, quantities are normalized by the best on-policy result from Ray et al. (2019) for the given environmental configuration.

## B UNCONSTRAINED OFF-POLICY ACTOR-CRITIC, SQUASHED AND REGULARIZED

Below is pseudocode for the unconstrained version of our method, OPAC$^2$. It represents a simplification of the constrained approach provided in the main text.

---

**Algorithm 2** Off-Policy Actor-Critic, SQUAshed and REgularizeD (OPAC$^2$)

---

1: **Input:** Initial policy parameters $\theta$; $Q$ parameters $\phi$; $V$ parameters $\psi$
2: **Input:** Initial entropy weight $\alpha$
3: Set $V$ target equal to main parameters: $\psi_{\text{targ}} \leftarrow \psi$
4: Initialize optimizers with learning rates $\lambda_\pi = \lambda_V = \lambda_Q; \lambda_\alpha$
5: Initialize replay buffer $\mathcal{D} = \emptyset$, a ring buffer of fixed size
6: Initialize environment : $\mathbf{s} \sim p(\mathbf{s}_0)$
7: **for** iteration $k \in [0, \ldots, K-1]$ **do**
8:     **for** step $s \in [0, \ldots, S-1]$ **do**                    $\triangleright$ Typically take just one step ($S = 1$)
9:         Sample $\mathbf{a} \sim \pi_{\theta_k}(\mathbf{a}|\mathbf{s})$; observe $\mathbf{s}' \sim p(\mathbf{s}'|\mathbf{s}, \mathbf{a})$           $\triangleright$ One step in MDP
10:         $\mathcal{D} \leftarrow \mathcal{D} \cup \{(\mathbf{s}, \mathbf{a}, \mathbf{s}', r(\mathbf{s}, \mathbf{a}))\}$                    $\triangleright$ Update buffer
11:     **end for**
12:     **for** gradient step $g \in [0, \ldots, G-1]$ **do**
13:         Sample batch $B = \{(\mathbf{s}_i, \mathbf{a}_i, \mathbf{s}'_i, r_i, d_i)\}$ from $\mathcal{D}$           $\triangleright$ or, a batch for each parameter set
14:         Compute $Q$ error: $\mathcal{E}_Q(B) = \sum_i [Q(\mathbf{s}_i, \mathbf{a}_i) - (r_i + \gamma(1 - d_i)V_{\text{targ}}(\mathbf{s}_i))]^2$
15:         Update $Q$: $\phi \leftarrow \phi - \lambda_\phi \nabla_\phi \mathcal{E}_Q(B)$
16:         Sample $\mathbf{a}_{i,\pi} \sim \pi(\mathbf{a}|\mathbf{s}_i)$, using $\tanh$ squashing
17:         Compute $V$ error: $\mathcal{E}_V(B) = \sum_i [(V(\mathbf{s}_i) - Q(\mathbf{s}_i, \mathbf{a}_{i,\pi})]^2$
18:         Update $V$: $\psi \leftarrow \psi - \lambda_\psi \nabla_\psi \mathcal{E}_V(B)$
19:         Compute $A(\mathbf{s}_i, \mathbf{a}_{i,\pi}) = Q(\mathbf{s}_i, \mathbf{a}_{i,\pi}) - V(\mathbf{s}_i)$                    $\triangleright$ Normalize
20:         Sample $\mathbf{a}_{\text{rp},\pi} \sim \pi(\mathbf{a}|\mathbf{s}_i)$ with reparameterization trick, for $\tanh$ squashing
21:         Compute $\pi$ loss: $\mathcal{E}_\pi(B) = \sum_i [\alpha \log \pi_\theta(\mathbf{a}_{\text{rp},\pi}|\mathbf{s}_i) - A(\mathbf{s}_i, \mathbf{a}_{i,\pi}) \log(\pi_\theta(\mathbf{a}_{i,\pi}|\mathbf{s}_i)]$
22:         Update $\pi$: $\theta \leftarrow \theta - \lambda_\theta \nabla_\theta \mathcal{E}_\pi(B)$
23:         Compute $\alpha$ loss: $\mathcal{E}_\alpha(B) = -\alpha [\log(\pi(\mathbf{a}_{i,\pi}|\mathbf{s}_i)) + \mathcal{H}_{\text{target}}]$
24:         Update $\alpha$ : $\alpha \leftarrow \alpha - \lambda_\alpha \nabla_\alpha \mathcal{E}_\alpha(B)$
25:         Update value target: $\psi_{\text{targ}} \leftarrow \rho \psi_{\text{targ}} + (1 - \rho)\psi$
26:     **end for**
27: **end for**

---

## C  EXPERIMENTAL DETAILS

Full source code is available in the supplementary material, and will be released publicly pending review. Hyperparameters are listed in Table 1. For all Safety Gym experiments, we used a learning rate of $10^{-4}$ for all algorithms (matching the setting in the OpenAI `safety-starter-agents` accompanying the suite). For all DeepMind Control experiments, we used a learning rate of $3 \times 10^{-4}$ for SAC and TD3, following standard practice. We retained the learning rate of $10^{-4}$ for OPAC$^2$ on DeepMind Control. All methods shared the same learning rate for the logarithm of the entropy weight $\alpha$ ($5 \times 10^{-4}$) and all constrained experiments used a learning rate of $5 \times 10^{-6}$ was used for the Lagrangian $\beta$. For all DeepMind Control experiments, we configured OPAC$^2$ to use max-entropy regularization, finding that it performed better than the entropy bonus we used for experiments with mixed-sign rewards.

Throughout, all neural networks considered were multilayer perceptrons, with two hidden layers of 256 units each. As is standard, ReLU activations were used for SAC and TD3. We chose $\tanh$ activations for OPAC$^2$ in order to match on-policy methods with a similar policy update. We evaluated $\tanh$ activations on SAC and TD3 as well, but found them to make little difference. For experiments involving resetting, we followed the practice of Nikishin et al. (2022) of resetting all networks and optimizers (except for those corresponding to the learned temperature) every 200k environment steps. All traces shown reflect five random seeds.

The policy networks output the mean values of a multivariate normal distribution with diagonal covariance. For OPAC$^2$, control variances were optimized. Variances were independent of state for all constrained experiments with the Car and Point robots, as well as for unconstrained experiments for Car and Point on Goal and Push. They varied with state for all experiments with Doggo robot, as well as for the Car and Point robots on the unconstrained Button task. For SAC, control variances always varied with state.

| Parameter | Value |
|---|---|
| Discount | 0.99 |
| Replay Buffer Size | $10^6$ |
| Optimizer | Adam (Kingma & Ba, 2015) |
| Network Layers | 2 |
| Network Hidden Units (per layer) | 256 |
| Batch Size | 256 |
| Target Network Update Interval | 1 |
| $\tau$ (target network averaging) | 0.995 |
| Initial Exploration | 10000 |

Table 1: Shared hyperparameters for SAC, TD3, and OPAC[2]

As mentioned in Section 5 of the main text, we chose to evaluate our approach using the OpenAI Safety Gym (Ray et al., 2019). This choice was governed by our desire to test in conditions with clear cost-incentive trade-offs, significant stochasticity, adequate complexity, and available benchmarks.

The environments chosen were the most obstacle-rich of the publicly available environments; we considered all robots and all tasks. The Point robot is constrained to the 2D plane and has two control dimensions: one for moving forward/backward and one for turning. The Car robot also has two control dimensions, corresponding to independently actuated parallel wheels. It has a freely rotating wheel and, while it is not constrained to the 2D plane, typically remains in it. The Doggo robot is a quadrupedal robot with bilateral symmetry and 12 control dimensions. Several types of obstacles and tasks were present in the environments we evaluated. In all cases, the robot is given a fixed amount of time (1000 steps) to complete the prescribed task as many times as possible and is motivated by both sparse and dense reward contributions. In the "Goal" environments, the robot must navigate to a series of randomly-assigned goal positions, with a new target being assigned as soon as a goal is reached. In the "Button" environments, the robot must reach and press a sequence of goal buttons while avoiding other buttons. In the "Push" task, the robot must push a box to a series of goal positions. The set of obstacles are different for each task; among the three environments there are a total of five different constraint elements (hazards, vases, incorrect buttons, pillars, and gremlins), each with different dynamics. See Ray et al. (2019) for further details.

All of our experiments used a single indicator for overall cost at each time step (the OpenAI default). In the unconstrained experiments, each cost event (robot contacting obstacle) was assigned a fixed (negative) weight in the reward function. The Car and Point robots used large penalty weights that matched Markowitz et al. (2023); the small ones were reduced by a factor of 2. The DoggoButton2 environment used factors of 0.0125 and 0.00625 for large and small penalty weights, respectively, while DoggoGoal2 used 0.025 and 0.0125.

## D    ADDITIONAL UNCONSTRAINED RESULTS: SAFETY GYM

Below we supplement the unconstrained results for Safety Gym provided in the main text. We include all learning curves, as well as representative plots of TD error and Q approximation error, both on held-out validation transitions. Five episodes (5000 transitions) of evaluation data were collected every 10000 training steps throughout the learning process.

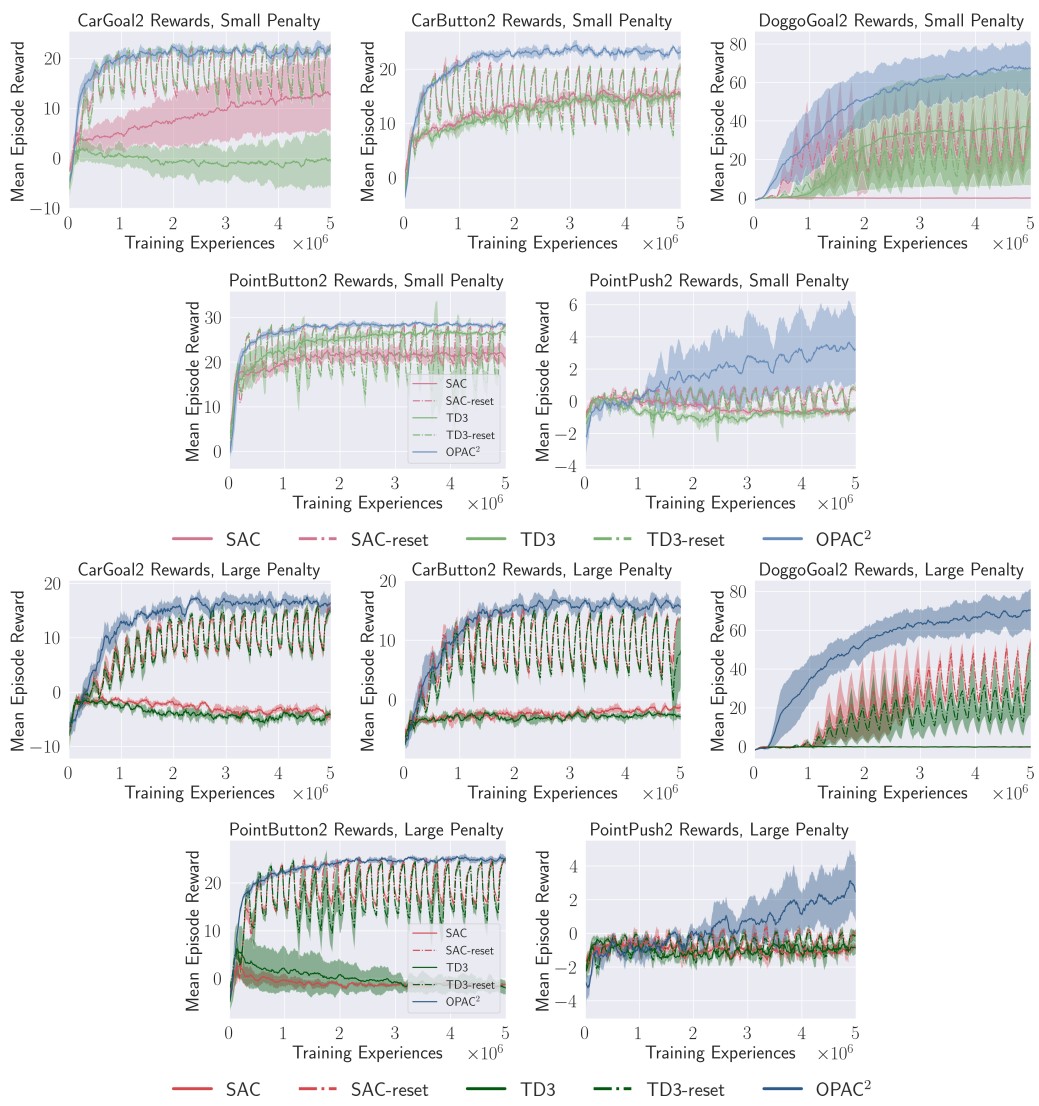

Figure 8: Unconstrained Learning on Safety Gym. SAC and TD3 struggle to learn without resetting. Even with periodic resets, they do not match the performance of OPAC$^2$.

As mentioned in the main text, we observed negative $Q$ errors in all Point and Car environments, while the trend was more variable with Doggo. While the magnitude of TD and Q errors varied per-environment, qualitative trends were consistent across environments.

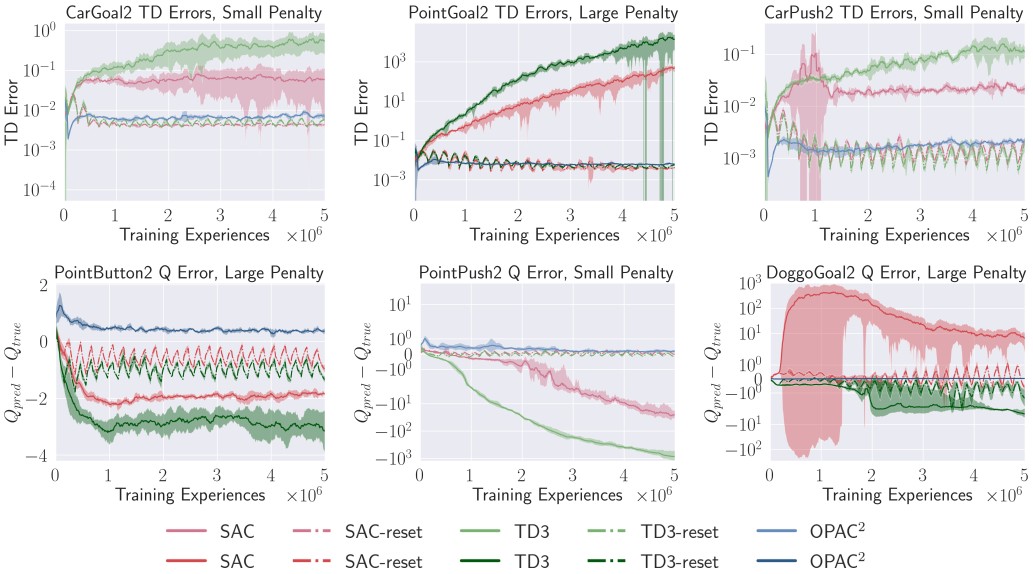

Figure 9: Validation TD error, Q estimation error plots for sample environments.

## D.1 Resetting with the Off-Policy Actor-Critic

We additionally explored the effect of resetting on OPAC$^2$. We found resetting to not be beneficial, and in fact typically be detrimental, to the learning of OPAC$^2$ in Safety Gym. This makes sense, considering that OPAC$^2$ has fairly accurate value estimates to begin with. Any small gains in value estimation accuracy provided by resetting are not sufficient to overcome its disruptive effects on network training.

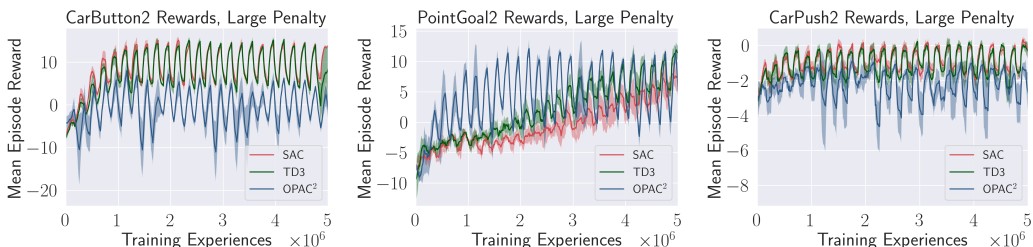

Figure 10: Resetting is not found to be beneficial for OPAC$^2$, and in fact is typically detrimental.

## D.2 Entropy Regularization Strategies

Below we plot the effect of different entropy regularization strategies in the DoggoButton2 and DoggoGoal2 environments. While not shown here, we found these trends to be consistent in the CarButton2 and PointButton2 environments.

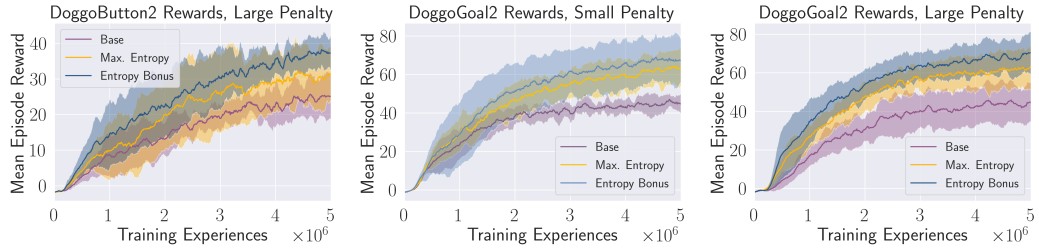

Figure 11: Different entropy regularization strategies on Doggo environments. Our entropy bonus consistently, though sometimes narrowly, outperforms maximum entropy and no entropy regularization in these environments.

### D.3 JUSTIFYING TWO $Q_c$ NETWORKS

We found 2 $Q_c$ networks to provide enhanced constrained learning for both SAC and TD3. We used this for the results in the main text in order to give the baselines the best chance at performing well. While the gap was larger in some environments than others, we always observed at least some improvement through the addition of an extra $Q_c$ network, for both algorithms.

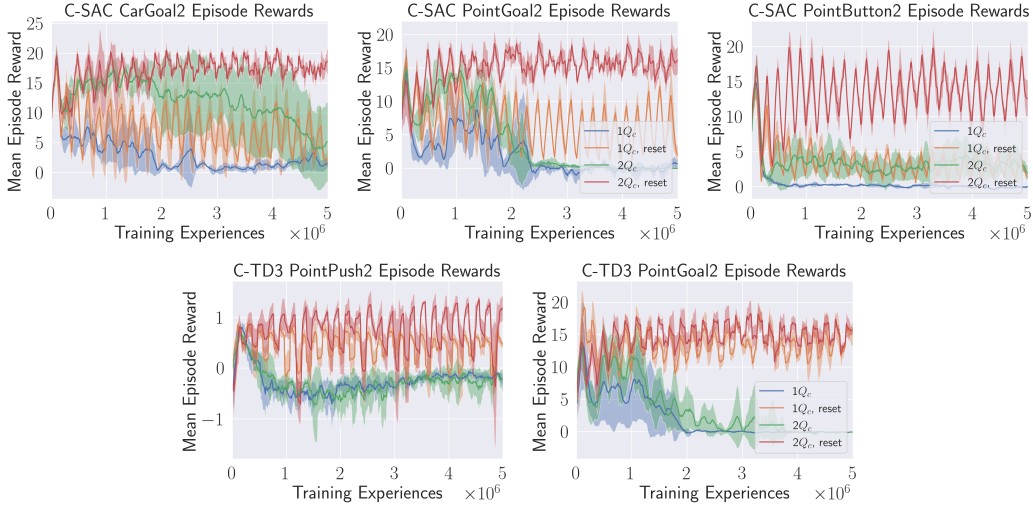

Figure 12: Comparison of positive reward accumulation in constrained learning. In all cases, cost plots were qualitatively similar. **Top row**: comparison using SAC. **Bottom row**: comparison using TD3.

## E ADDITIONAL CONSTRAINED RESULTS: SAFETY GYM

Here we include additional reward and cost results from Safety Gym. As mentioned in the main text, OPAC[2] was able to match specified cost levels in every environment tested, while providing higher positive rewards than other methods.

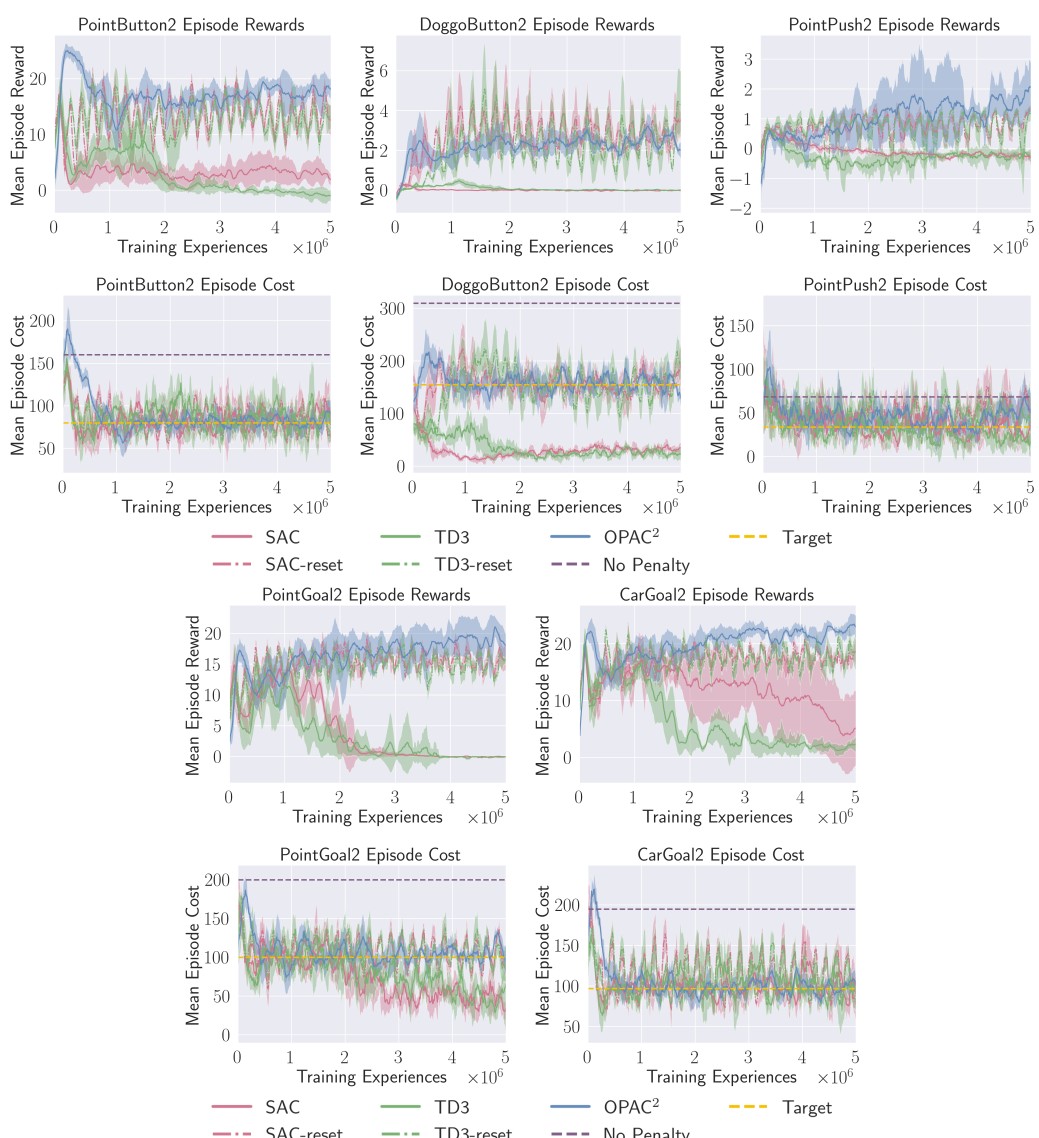

Figure 13: Additional results from constrained learning experiments from Safety Gym. **Top rows**: reward accumulated by agents. **Bottom rows**: cost converges to the target level (yellow). Note that reward accumulation initially increases sharply while $\beta$, the penalty weighting, is low. It dips and then rises again as performance is optimized at the appropriate cost level.

We additionally provide a few extra plots reflecting the superior ability of OPAC$^2$ to mitigate validation TD error for both cost and reward in constrained learning, compared to SAC and TD3.

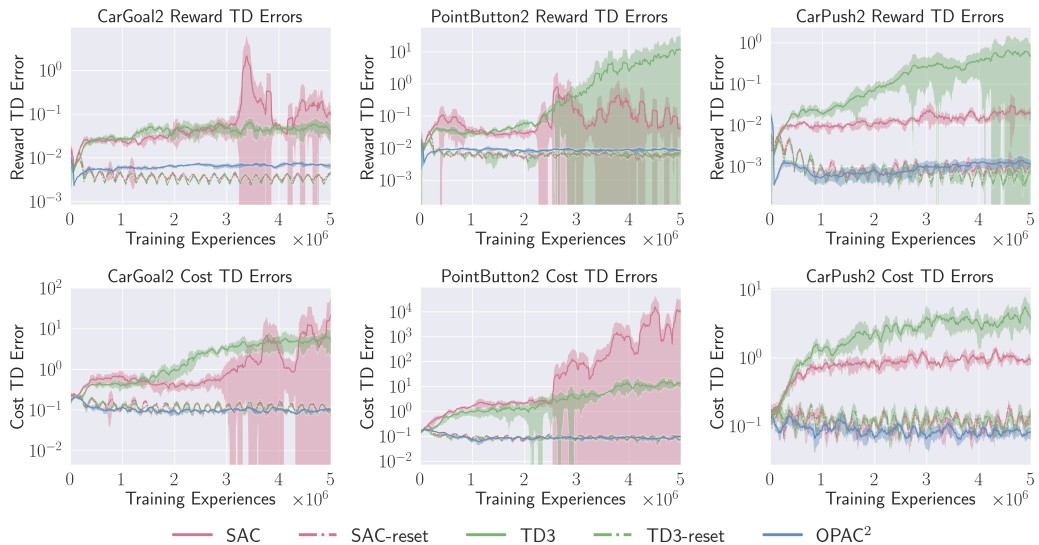

Figure 14: **Top row**: Validation TD error for reward in constrained learning. **Bottom row**: Validation TD error for cost in constrained learning.

# F  ADDITIONAL UNCONSTRAINED RESULTS: DEEPMIND CONTROL

For completeness, we include the learning curves for our evaluation of OPAC$^2$, SAC, and TD3 in each of the 10 DeepMind Control Suite environments we tested. We also provide an overall average plot, across the 10 environments.

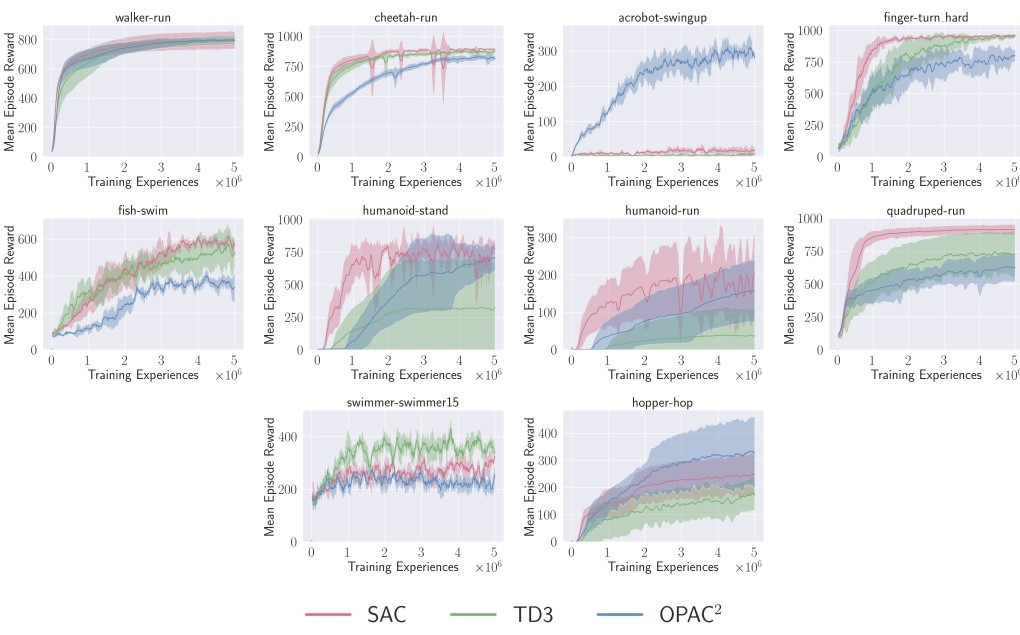

Figure 15: Per-environment results for 10 tasks from the DeepMind Control Suite.

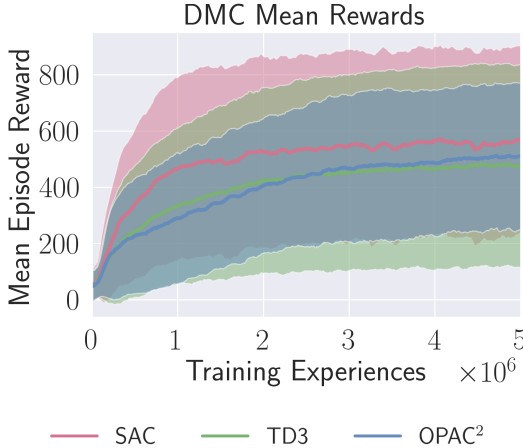

Figure 16: Overall average performance of SAC, TD3, and OPAC$^2$ on 10 tasks from the DeepMind control suite.

