# OpenReview forum: "Handling Cost and Constraints with Off-Policy Deep Reinforcement Learning"
_ICLR.cc/2024/Conference — Submitted to ICLR 2024_

### Official Review · Reviewer_SRMH · 2023-10-28

**Soundness:** 3 good
**Presentation:** 2 fair
**Contribution:** 2 fair
**Rating:** 3
**Confidence:** 3

**Summary:**

This paper considers the RL problem with mixed-signed reward functions, where there are costs in the MDP steps. The authors proposes C-OPAC2 method to solve the problem. At the same time, the author suggests useful tricks including resetting $Q$ and policy network and  remove the maximization term. The superiority of the method is validated from the robotic navigation task.

**Strengths:**

1. The paper considers an important problem in RL community, where costs exists are constraints are required to satisfy.
2. The experiment studies demonstrate great potential of the proposed method.

**Weaknesses:**

1. Although the studied problem is interesting, the manuscript is not written and hard to follow.
2. Some notations in Algorithm 1 are not mentioned in Section 3 and Section4, we suggest the authors to explain some important notations in Section 3&4.
2. For mixed-signed reward, it is suggested to give more practical examples, otherwise the statements seems confusing. The authors first claim the mixed-signed reward, later in equation 8, the $c(t)$ is cost. Then in Section 4.2.1, the $C(\tau)$ is constraint. So the constraint could be part of negative reward?
3. In abstract, the authors mentions removing maximization term and reseting the networks, however, this seems not the crucial part of the manuscript.

**Questions:**

1. The authors considers the constrained RL problem, and it is suggested to consider and compare some related reference [1]
2. In Section 3, the author mentions ``'When Q is underestimated, |$Q_c$| will tend to be overestimated and |$Q_r$| underestimated,'. Why will  |$Q_c$| will tend to be overestimated and |$Q_r$| underestimated, could the author give more insights or theoretical explanations? Why not both Q networks underestimated?
3. There are some concerns why we use two Q networks for $Q_r$ and $Q_c$, even if the reward are assigned at the same time, we can still use one network to estimate.
The reviewer will consider the increase the rating when the concerns are fully resolved.

[1]: Reward Constrained Policy Optimization. https://arxiv.org/abs/1805.11074

---

> ### Author Response · Authors · 2023-11-18
> **Review Response**
>
> Thank you for your review.  We believe that addressing the points you raised has helped us to significantly improve the clarity of the manuscript. Below are responses to the weaknesses you noted and the questions you had. We hope that our edits and these responses resolve the issues you mentioned, but please let us know if you have any outstanding concerns.
>
> Weaknesses
>
> - We agree with the criticism that the initial version of the paper was hard to follow, and have made numerous changes to correct this. In particular, we edited Section 3, as it was not clearly written and seemed to be a general source of confusion.
>
> - We have revised Sections 2-4 as well as Algorithm 1 in an effort to make them clearer and to ensure that all notation is defined.
>
> - We have added further examples of mixed-sign rewards to the text. These range from robotics (navigation in the presence of obstacles, robotic surgery) to resource allocation (when both performance and efficiency must be considered), to financial decision-making (where the risk of losses exists in the pursuit of gains).
>
> - We have tried to clarify what we mean by cost, and in fact removed its definition in Section 3 to help with this. Cost, $c(t)$, is provided separately from reward at each time step of a constrained Markov decision process (CMDP). The constraint $C(\tau)$ is a function of the costs $c(1)… c(T)$ over the trajectory $\tau \equiv \mathbf{s}_1, \mathbf{a}_1, \ldots,  \mathbf{s}_T, \mathbf{a}_T $.
>
> - In our case,  $C(\tau)$ was simply the total cost accumulated by the agent over the trajectory: $C(\tau) = \sum_{t=1}^T c(t)$.
>
> - We have updated the abstract in an effort to make the contribution more clear. In particular, we never meant to imply that the use of resetting in conjunction with SAC and TD3 is novel. Rather, it was included as a way to study and address the issue we observed with mixed-sign rewards, as well as a comparison point for our approach (OPAC$^2$).
>
> Questions
>
> - We did in fact compare our approach with existing constrained RL methods, including Reward Constrained Policy Optimization (RCPO). We apologize that this was not made clear. More specifically, we compared our constrained results with those reported in [1], which leveraged RCPO in conjunction with PPO, TRPO, and a risk-sensitive approach to produce state-of-the-art on-policy constrained learning in Safety Gym. FOCOPS [2] was also included in that work. We used the same environments and cost targets as [1], achieving performance very similar to the best results there with something like 50 times less data.
>
> - We agree that our explanation in Section 3 was inadequate, and in fact significantly edited it to better explain why bias in $Q$ estimation is likely to impact competing objectives asymmetrically. The key insight is that, as previously reported [3], agents trained off-policy and with objectives that maximize $Q$ will tend to overemphasize experiences from early in training. When multiple terms are present, this amounts to fixation on the reward terms that are most easily accessed by an agent of low capability. The issue becomes acute when terms explicitly compete, as in environments with mixed-sign rewards.
>
> -  We see how the way we wrote Section 3 could lead to misunderstandings of how $Q$ estimation of rewards and costs was handled. We have edited the paper to resolve this issue; here is a summary of how it is handled in our algorithms. For unconstrained learning, OPAC$^2$ requires one $Q$ function, one $V$ function, and the policy $\pi$. In removing a $Q$ function and adding $V$, our approach maintains the same number of networks and has slightly fewer parameters than SAC and TD3 (since the input to $V$ is $\mathbf{s}$ rather than $(\mathbf{s},\mathbf{a})$). For constrained learning, we have two $Q$ functions, two $V$ functions, and a policy $\pi$. The two $Q$ and two $V$ are necessitated by separate handling of positive rewards and costs in a CMDP and the changing scaling factor between them ($\beta$). Again, we have the same number of networks and fewer parameters than the constrained versions of SAC and TD3.
>
> [1] Markowitz et al. A Risk-Sensitive Approach to Policy Optimization.  AAAI; https://arxiv.org/abs/2208.09106. 2023.
>
> [2] Zhang et al. First Order Constrained Optimization in Policy Space. NeurIPS; https://arxiv.org/abs/2002.06506. 2020.
>
> [3] Nikishin et al. The Primacy Bias in Deep Reinforcement Learning. ICML; https://arxiv.org/abs/2205.07802. 2022.

---

### Official Review · Reviewer_FTtw · 2023-10-29

**Soundness:** 2 fair
**Presentation:** 2 fair
**Contribution:** 2 fair
**Rating:** 5
**Confidence:** 2

**Summary:**

This study focuses on environments with "mixed-sign" reward functions. The authors investigate the root cause of the poor performance and discover that it arises from an asymmetric error in estimating the magnitude of returns associated with terms of different signs. And provide a novel algorithm (OPAC^2) building around the off-policy actor-critic. The experiments demonstrate that the proposed algorithm outperforms state-of-the-art methods augmented by resetting.

**Strengths:**

1. The proposed method is detailed and easy to follow.
2. The proposed algorithm seems competitive in both constrained and unconstrained settings.

**Weaknesses:**

1. The authors analyze why mixed-sign rewards are problematic in Sec.3, which is not so obvious to me. Could you please provide some more pieces of evidence to help me understand this?
2. I'm not quite sure why resetting can improve this issue. Do you have any intuition on this?
3. Similarly, I don't quite understand the intuition behind OPAC. Are there any ablation studies available?
4. Do you compare OPAC with other Constrained RL algorithms?

**Questions:**

Please refer to Weaknesses

---

> ### Author Response · Authors · 2023-11-18
> **Review Response**
>
> Thank you for your review.  We believe that addressing the points you raised has helped us to significantly improve the clarity of the manuscript. Below are responses to the weaknesses you noted.  We hope that our edits and these responses resolve the issues you mentioned, but please let us know if you have any outstanding concerns.
>
> - We agree that Section 3 was not written clearly, and have edited it to better explain why bias in $Q$ estimation is likely to impact competing objectives asymmetrically.  The key insight is that, as previously reported [1], agents trained off-policy and with objectives that maximize $Q$ will tend to overemphasize experiences from early in training. When multiple terms are present, this amounts to fixation on the reward terms that are most easily accessed by an agent of low capability. The issue becomes acute when terms explicitly compete, as in environments with mixed-sign rewards.
>
> - Regarding why resetting helps with this problem: as noted in [2], resets combat learning failure due to inaccurate value estimates. Intuitively, they combat overfitting to experiences collected early in training. Indeed, we found (Figure 2 and Appendix D) that resetting significantly decreased both TD error and $Q$ estimation error in validation, leading to better agent performance. OPAC$^2$ achieves superior agent performance without resetting because it is not as inherently prone to inaccurate value estimates.
>
> - Regarding the intuition behind OPAC$^2$: there are a few elements that come together, which can be thought of in terms of bias and variance tradeoffs. In replacing the policy update of SAC (which maximizes Q via the reparameterization trick) with the "gentler" OPAC update, we largely remove the systematic bias on $Q$ estimates that is problematic in environments with competing objectives. The policy update in SAC is provably lower variance than the basic off-policy actor-critic, though this gap is closed through the use of a state-dependent baseline (i.e., value function). Our approach also leverages the ``squashing” trick of SAC to remove bias due to action clipping associated with control bounds, and may employ different forms of entropy regularization to enhance exploration.  Our experiments demonstrate that we are in fact able to build a practical algorithm around the policy update of the off-policy actor-critic; that is, in environments with mixed-sign rewards, the benefits of the superior value estimation of OPAC$^2$ more than outweigh the downsides of the potentially higher variance of its policy updates.
>
> - Regarding comparison of OPAC$^2$ with other constrained RL algorithms: yes, we do. Given that multiple reviewers missed this, we must apologize for not making this clear. See Section 5.2 of the revised manuscript. In addition to outperforming constrained SAC and TD3 in every environment tested, our approach essentially matches the performance of state-of-the-art on-policy approaches \emph{using roughly 50 times less data}. This latter statement is based on [3], where the same Car and Point environments were tested with the same cost thresholds.
>
> [1] Nikishin et al. The Primacy Bias in Deep Reinforcement Learning. ICML; https://arxiv.org/abs/2205.07802. 2022.
>
> [2] Li et al. Efficient Deep Reinforcement Learning Requires Regulating Overfitting. ICLR; https://arxiv.org/abs/2304.10466. 2023.
>
> [3] Markowitz et al. A Risk-Sensitive Approach to Policy Optimization.  AAAI; https://arxiv.org/abs/2208.09106. 2023.

---

### Official Review · Reviewer_mpXe · 2023-11-01

**Soundness:** 2 fair
**Presentation:** 3 good
**Contribution:** 2 fair
**Rating:** 3
**Confidence:** 4

**Summary:**

The paper is about RL with "mixed-sign" rewards, i.e., the reward function includes independent incentive and cost terms. The authors then argue that learning based on maximizing the summation of two Q-function approximations would lead to overestimates. Relying on these observations, the authors explore two approaches based on resetting and an off-policy actor-critic that does not include Q maximization in the policy improvement step. Experiments are conducted using OpenAI SafetyGyms to compare these new approaches with the standard SAC and TD algorithms.

**Strengths:**

The problem of solving RL with mixed-sign rewards is relevant and worthy of attention. The exploration based on decomposing the Q and V network, and resetting, seems to be a good approach. Experiments show that the new algorithm performs well compared to SAC and TD. The proposed algorithm is not difficult to implement.

**Weaknesses:**

I believe the paper's contributions are somewhat incremental and unclear at several points. Please find my comments below:

- I assume the main selling point of the work is Section 3, where the authors discuss the limitations of learning two Q-functions. This section, however, is not clear and not convincing. First, the authors state that under mixed-sign rewards, the Q function can be decomposed into two Q functions, one for the rewards and one for the costs. Why should this decomposition be considered and analyzed? Why shouldn't we keep the overall Q function and learn it based on the total reward $r_{total}$? I can see that the decomposition would lead to over or underestimations for both the rewards and the costs, as the summation would fail to manage how the costs and rewards contribute to the overall rewards. So, it may not be a suitable approach to handle mixed-sign reward situations.
- Later on, Algorithm 1 is also based on two Q and two V functions. This raises the question of how this way of learning compares to learning one Q function based on $r_{total}$. In other words, can we say anything about the equivalence or convergence of this decomposition approach compared to single-Q learning?
- The constrained approach requires introducing the threshold $d$." I am not sure how this can be done systematically as all we know are only the costs and rewards and an upper bound on the accumulated costs is essentially not available. In the experiments, the authors say that "we chose a target cost level equal to half the cost accumulated by a fully-trained TRPO agent unaware of cost." It is very unclear why it should be chosen this way. This needs more justifications.
- The constrained RL approach based on Lagrange multipliers is simply to convert the overall rewards from $r+c$ to $r+\beta c$. This does not seem to be a good approach.
- The Resetting approach is clearly not new. The authors simply apply this to their problem context and find improvements. So it should not be considered a major contribution

**Questions:**

Please see the Weaknesses above

---

> ### Author Response · Authors · 2023-11-18
> **Review Response**
>
> Thank you for your review. We believe that addressing the points you raised has helped us to significantly improve the clarity of the manuscript. Below is a brief clarification and responses to the weaknesses you noted. We hope that our edits and these responses resolve the issues you mentioned, but please let us know if you have any outstanding concerns.
>
> - First, we want to provide a clarification about an item in your summary: “The authors then argue that learning based on maximizing the summation of two $Q$-function approximations would lead to overestimates.” In the unconstrained setting, we don’t actually learn two $Q$-functions and sum them; we only learn one $Q$-function. We do learn different $Q$ functions for incentives and costs in the constrained setting, as well as a weighting factor for the cost term relative to the incentive term. The confusion here likely originates from our initial version of Section 3, which we agree was not written clearly. This section has been revised in the new draft (see next point).
>
> - We see the key contributions of this work being (1) the identification of the issue in Section 3 and (2) the proposal of OPAC$^2$ to address it. We rewrote Section 3 so as to better explain why bias in $Q$ estimation is likely to impact competing objectives asymmetrically. The key insight is that, as previously reported [1], agents trained off-policy and with objectives that maximize $Q$ will tend to overemphasize experiences from early in training. When multiple terms are present, this amounts to fixation on the reward terms that are most easily accessed by an agent of low capability. The issue becomes acute when terms explicitly compete, as in environments with mixed-sign rewards. The decomposition originally included in Section 3 was meant simply to illustrate the implication of independent terms in the reward function; in unconstrained learning we only ever learn one $Q$ function.
>
> - For unconstrained learning, OPAC$^2$ requires one $Q$ function, one $V$ function, and the policy $\pi$. In removing a $Q$ function and adding $V$, our approach maintains the same number of networks and has slightly fewer parameters than SAC and TD3 (since the input to $V$ is $\mathbf{s}$ rather than $(\mathbf{s},\mathbf{a})$). For constrained learning, we have two $Q$ functions, two $V$ functions, and a policy $\pi$. The two $Q$ and two $V$ are necessitated by separate handling of positive rewards and costs in a CMDP and the changing scaling factor between them ($\beta$).  Again, we have the same number of networks and fewer parameters than the constrained versions of SAC and TD3. We do see how these points were unclear in the initial draft, and have tried to improve the clarity in our revision.
>
> - Regarding your comment requesting justification on the choice of cost threshold $d$: we should have included the rationale initially but have now added it to the draft. The intention was to provide a level that required the agent to strongly consider safety, but not be fully deterred from pursuing goals.  This choice also had the benefit of matching the thresholds used in [2], allowing us to directly compare the performance of our approach with that of state-of-the-art on-policy methods.
>
> - Regarding your comment on the merits of Lagrangian approaches: we do agree that they are suboptimal in that they do not enforce safety during training. However, we find them to be effective in allowing the final agent to maximize positive rewards subject to an acceptable level of average cost.   This is still a valuable capability, and removes the need for reward shaping. Our formulation ties directly to the cost accumulated by the current agent (calculating the constraint over the most recent epoch in the replay buffer), allowing for an interpretable notion of safety.
>
> - We want to be clear that we do not consider our use of resetting to be a major contribution, and have tried to make this point clearer in the revised draft. We consider the observation that resetting helps SAC and TD3 significantly in environments with mixed-sign rewards to be interesting; however, the results with resetting are more intended to provide a comparison point for our OPAC$^2$ approach.
>
> [1] Nikishin et al. The Primacy Bias in Deep Reinforcement Learning. ICML; https://arxiv.org/abs/2205.07802. 2022.
>
> [2] Markowitz et al. A Risk-Sensitive Approach to Policy Optimization.  AAAI; https://arxiv.org/abs/2208.09106. 2023.

---

### Official Review · Reviewer_VQxc · 2023-11-09

**Soundness:** 2 fair
**Presentation:** 2 fair
**Contribution:** 2 fair
**Rating:** 3
**Confidence:** 4

**Summary:**

This paper explores the challenges of reinforcement learning (RL) in settings where the environment returns both positive and negative rewards. The authors highlight the limitations of standard off-policy RL algorithms such as SAC and TD3 in handling such scenarios and discuss methods like periodic network resets and constrained-MDPs to mitigate these issues. They propose a method called, constrained off-policy actor-critic algorithm that combines elements of said approaches to build a method that can work in such scenarios. The performance of their method is evaluated using some of the OpenAI Safety Gym benchmark tasks.

**Strengths:**

The paper effectively motivates the difficulties of employing standard RL algorithms in scenarios where the environment provides both positive and negative rewards. The writing is clear and engaging with a well-structured flow up to section 4.2.1.

**Weaknesses:**

- This is an empirical paper as the proposed method is nothing but the combination and examination of existing ideas without introducing new ones. Being an empirical paper is not a negative point, however, it requires comprehensive and thorough results. Unfortunately, it is not the case in this paper. In particular, the paper uses OpenAI Safety Gym benchmark to evaluate their method but failed to include more Safe-RL methods, like [1], etc. It was shown in previous papers that CMDP methods work the best in this benchmark and using standard RL methods ( MDP-based not CMDP) don't result in good performance.


- The authors' main contribution appears to be Algorithm 1, which is briefly described in Section 4.2.1. There are many issues here. First of all, this algorithm has many moving parts and is a rather very complicated method. For instance, it requires assigning 5 different learning rates (i.e. $\lambda_\beta, \lambda_\phi, \lambda_\psi, \lambda_\theta, \lambda_\alpha$) which clearly shows level of complexity in this method. In addition, while this method sometimes shows some improvement in some of the benchmarks, it remains unclear what drives these enhancements. It's crucial to note that the proposed algorithm is evaluated against not right baselines which are not designed for this specific problem setting. The results are also mixed and this method doesn't show a consistent trend in the experiments. For instance, compare results of DoggoGoal and CarPush in Figure 2. Finally, writing of section 4.2.1 and experiments section need major work as it's either too shallow (e.g. 4.2.1) or excessively and unnecessarily detailed, making it challenging to follow ( e.g. especially in the experiment section).

-  The idea of having an environment that returns multiple rewards is a valid idea. However, limiting it to just positive and negative rewards seems narrow. This scenario appears more akin to a constrained markov decision process (CMDP), where one function serves as a reward, and the other as a cost. Authors could have studied this topic in a multi-objective RL setting where there are multiple rewards and the goal is to find a policy that is "optimal" across different rewards. This is a well-studied topic ( see [2], etc) and this paper seems to have selected a setting which is very limited.

Despite this paper studies an important problem, unfortunately, it presents several shortcomings as mentioned above and is not yet ready for ICLR at the current form and requires major work.

[1] Conservative Safety Critic, https://arxiv.org/abs/2010.14497

[2] A Distributional View on Multi-Objective Policy Optimization https://arxiv.org/abs/2005.07513

**Questions:**

In page 4, it is mentioned that "When the reward function has independent terms of different signs, errors in the magnitude of the estimates for |Qr| and |Qc| will grow in opposite directions". This might be true in very limited cases, but I don't think that always holds. Do you have any mathematical or numerical evidence that justify your claim?

---

> ### Author Response · Authors · 2023-11-18
> **Review Response (1/2)**
>
> Thank you for your review. We believe that addressing the points you raised has helped us to significantly improve the clarity of the manuscript. Here are responses to the weaknesses you noted as well as your question. We hope that our edits and these responses resolve the issues you mentioned, but please let us know if you have any outstanding concerns.
>
> Weaknesses
>
> - Regarding the point that CMDP methods work best on Safety Gym:  We propose and analyze two variants of our OPAC$^2$ algorithm: one in the unconstrained setting and one in the constrained setting, comparing each to standard baselines in their respective settings. We compare the unconstrained version of our algorithm with SAC and TD3, and compare the constrained version of our algorithm with constrained versions of SAC and TD3; in all cases configurations among the three were as similar as possible. We agree that previous unconstrained and constrained approaches either do not perform well or are extremely sample inefficient (for instance, [1]); hence our OPAC$^2$ results represent a major improvement in both the unconstrained and constrained settings.
>
> - Regarding the relation to the Conservative Safety Critic---we very much appreciate this reference and have added it to our Related Work section. While the safe exploration paradigm it considers is extremely important for real-world applications, it is not the setting we consider here. Our constrained approach is instead aligned with approaches like RCPO [2], which cannot guarantee safety during training but which can maximize positive reward accumulation given an acceptable leverage of average cost by the end of training. This still has utility, in that it removes the need for manual reward shaping. We have made this distinction clear in the revised text.
>
> - We agree that our pseudocode for Algorithm 1 appeared more complicated than needed, and have simplified our presentation accordingly. We also include the simpler unconstrained algorithm in the Appendix. We’d like to note a couple of things about complexity. First, we initially included all of the different learning rates in the paper in order to emphasize the flexibility available in tuning the method. However, our experiments did not require us to do excessive tuning. We simply chose $\lambda_\theta = \lambda_\psi = \lambda_\phi = 0.0001$, as in the SAC implementation that accompanied Safety Gym.  The default value of $\lambda_\alpha$ for SAC was used, and $\lambda_\beta$ was chosen to be significantly smaller than $\lambda_\theta$, as recommended by RCPO. Second, we note that our method does not require additional networks compared to SAC or TD3. OPAC$^2$ does add the value network but does not require a second $Q$ network; hence the total number of networks is the same and OPAC$^2$ actually uses slightly fewer parameters (because the input to $V$ is $\mathbf{s}$ rather than $(\mathbf{s},\mathbf{a})$).
>
> -  Regarding your comment “In addition, while this method sometimes shows some improvement in some of the benchmarks, it remains unclear what drives these enhancements”: we agree that our original explanation (Section 3) could have been written better, and have revised it. In terms of what drives the improved performance, we provide strong empirical evidence that this is due to removing the explicit maximization of the Q function present in SAC and TD3, which is subject to much stronger systematic estimation bias than our approach.
>
> - We respectfully disagree that the algorithm is not evaluated against the right baselines, or that the results are inconsistent. In all 8 Safety Gym environments tested, OPAC$^2$ outperforms SAC and TD3 (both with and without resetting). This is true with both small and large static penalties and when the problem is formulated in the constrained setting (24 experiments total).
>
> - We agree with your assessment on the length/depth of section 4.2.1 and the experiments section.  We have revised section 4.2.1 to include more detail (though we are still limited by space), and have edited the experiments section to be clearer and more concise.
>
> - We would like to provide a clarification regarding the point about “having an environment that returns multiple rewards.” In the unconstrained setting we study, the environment is only ever returning a single scalar reward. What we meant is that the value of that scalar reward is calculated as the sum of multiple reward terms, where each term might measure something different about the environment (some could represent “rewards” and some could represent “costs”). In the constrained setting, we do indeed have an environment return both a reward and a cost. We have revised the manuscript to make this clearer.

---

> > ### Author Response · Authors · 2023-11-18
> > **Review Response (2/2)**
> >
> > Questions
> >
> > - We agree that our statement “When the reward function has independent terms of different signs, errors in the magnitude of the estimates for $|Q_r|$ and $|Q_c|$ grow in different directions” was hard to follow. We have edited Section 3 to better explain why bias in $Q$ estimation is likely to impact competing objectives asymmetrically. The key insight is that, as previously reported [3], agents trained off-policy and with objectives that maximize $Q$ will tend to overemphasize experiences from early in training. When multiple terms are present, this amounts to fixation on the reward terms that are most easily accessed by an agent of low capability. The issue becomes acute when terms explicitly compete, as in environments with mixed-sign rewards.
> >
> > [1] Markowitz et al. A Risk-Sensitive Approach to Policy Optimization. AAAI; https://arxiv.org/abs/2208.09106. 2023.
> >
> > [2] Tessler et al. Reward Constrained Policy Optimization. ICLR; https://arxiv.org/abs/1805.11074. 2019.
> >
> > [3] Nikishin et al. The Primacy Bias in Deep Reinforcement Learning. ICML; https://arxiv.org/abs/2205.07802. 2022.

---

### Author Response · Authors · 2023-11-18
**General Response to Reviews**

Thanks to all of our reviewers for the time they spent reviewing our paper and the useful comments they provided. Given these responses, we have endeavored to significantly tighten the writing. We agree with the reviewers that some elements of the paper were not presented well initially, and hope that the revised submission addresses the issues raised.

We would like to call out a few aspects in this general post, in addition to responding in further detail to each individual reviewer below.

First, we would like to note that we edited section 3 in order to better explain why bias in $Q$ estimation is likely to impact competing objectives asymmetrically. The key insight is that, as previously reported [1], agents trained off-policy and with objectives that maximize $Q$ will tend to overemphasize experiences from early in training. When multiple terms are present, this amounts to fixation on the reward terms that are most easily accessed by an agent of low capability. The issue becomes acute when terms explicitly compete, as in environments with mixed-sign rewards.

Second, we would like to clarify the comparison of our method with other constrained approaches. In addition to outperforming constrained SAC and TD3, with and without resets, in every environment tested, our approach may be directly compared with state-of-the-art on-policy approaches. In [2], the same environments and cost thresholds were used as in our experiments here; constrained OPAC$^2$ roughly matches the best reported performance while using roughly 50 times less data.

Finally, since the original submission we have continued to experiment with OPAC$^2$ on environments from the DeepMind Control suite (DMC; [3]), where mixed-sign rewards are not present. We found that OPAC$^2$ could in fact be used to generate performance very competitive with SAC and TD3 in this setting, and in fact was the only algorithm of the three that produced competent learning in all 10 tested DMC environments. This attests to the general reliability of our approach and is discussed in the updated submission. We also added the DMC configuration files to our supplementary materials, in order to ensure reproducibility.

Thank you in advance for considering these edits.  We would appreciate if you could let us know of any outstanding concerns that remain or, if your concerns were addressed, improve your support for the paper.

[1] Nikishin et al. The Primacy Bias in Deep Reinforcement Learning. ICML; https://arxiv.org/abs/2205.07802. 2022.

[2] Markowitz et al. A Risk-Sensitive Approach to Policy Optimization. AAAI; https://arxiv.org/abs/2208.09106. 2023.

[3] Tassa et al. DeepMind Control Suite. https://arxiv.org/abs/1801.00690. 2018.

---

> ### Author Response · Authors · 2023-11-23
> **Final Version Posted**
>
> Thanks again to our reviewers for their time and feedback on this work.  We are posting to note that we made additional minor revisions from the rebuttal version of the paper that we submitted a few days ago.  We believe the contribution to be valuable, and appreciate your consideration of the updated submission.

---

### Meta-Review · Area_Chair_8i2G · 2023-12-06

**Metareview:**

While the reviewers agreed that topic addressed in the paper is interesting and important, there was consistent feedback that the presentation of the paper had to be significantly reworked. There was also feedback regarding experiments.

While the authors did claim to have made substantial changes to the paper, it wasn't clear when looking at the new pdf what was changed (differences were not highlighted) and it feels like the changes are too significant to be within the scope of the author/reviewer discussion |(from the call for paper "Area chairs and reviewers reserve the right to ignore changes that are significantly different from the original paper."). I think this paper falls in the case where there needs a new full round of reviews before being considered for acceptance.

**Justification For Why Not Higher Score:**

consistent feedback that the presentation of the paper made it not ready for publication.

**Justification For Why Not Lower Score:**

N/A

---

### Decision · Program_Chairs · 2024-01-16

Reject